# Genetic Determinants of Poor Response to Treatment in Severe Asthma

**DOI:** 10.3390/ijms22084251

**Published:** 2021-04-20

**Authors:** Ricardo G. Figueiredo, Ryan S. Costa, Camila A. Figueiredo, Alvaro A. Cruz

**Affiliations:** 1Pulmonology Division, Department of Health, Universidade Estadual de Feira de Santana, Feira de Santana 44036-900, Brazil; 2Fundação ProAR, Salvador 40000-000, Brazil; ryanscosta@yahoo.com.br (R.S.C.); cavfigueiredo@gmail.com (C.A.F.); cruz.proar@gmail.com (A.A.C.); 3Instituto de Ciências da Saúde, Universidade Federal da Bahia, Salvador 40170-110, Brazil; 4Faculdade de Medicina, Universidade Federal da Bahia, Salvador 40170-110, Brazil

**Keywords:** severe asthma, genetics, pharmacogenetics, precision medicine

## Abstract

Severe asthma is a multifactorial disorder with marked phenotypic heterogeneity and complex interactions between genetics and environmental risk factors, which could, at least in part, explain why during standard pharmacologic treatment, many patients remain poorly controlled and at an increased risk of airway remodeling and disease progression. The concept of “precision medicine” to better suit individual unique needs is an emerging trend in the management of chronic respiratory diseases. Over the past few years, Genome-Wide Association Studies (GWAS) have revealed novel pharmacogenetic variants related to responses to inhaled corticosteroids and the clinical efficacy of bronchodilators. Optimal clinical response to treatment may vary between racial/ethnic groups or individuals due to genetic differences. It is also plausible to assume that epigenetic factors play a key role in the modulation of gene expression patterns and inflammatory cytokines. Remarkably, specific genetic variants related to treatment effectiveness may indicate promising pathways for novel therapies in severe asthma. In this review, we provide a concise update of genetic determinants of poor response to treatment in severe asthma and future directions in the field.

## 1. Introduction

Asthma is a chronic respiratory disease usually characterized by inflammatory changes in the lower airways that requires long-term management. It has emerged as a major public health condition estimated to affect over than 339 million people worldwide and influenced by a puzzling interaction of environmental and genetic factors [1,2]. The rising prevalence of asthma in industrialized and developing countries over the past few decades has highlighted the importance of understanding the biological mechanisms that contribute to asthma development [3].

Asthma is a complex syndrome with a variable natural history and clinical development influenced by multiple genes and environmental exposures [2,4,5]. As illustrated in Figure 1, the severe asthma phenotype is expressed by diverse genetic predispositions that may favor abnormal pulmonary growth with lower values of lung function, lower respiratory tract infections in early life, bronchial hyperresponsiveness and airway inflammation, and delayed immune system maturation and allergic sensitization in childhood [4]. Moreover, phenotypic expression and the epigenetics pressure in some genes might have a significant influence on specific clusters of patients, for instance, in those with occupational asthma, smoking asthmatics, or those who were exposed to allergens in the first few years of life [6].

Severe asthma patients require close monitoring with typically challenging clinical management and a considerable financial burden. Despite pharmacologic treatment with anti-inflammatory and bronchodilators, a cluster of patients remain poorly controlled, and at increased risk of airway remodeling and disease progression [4,7].

The goal of understanding the genetic and epigenetic basis that affect the therapeutical response in severe asthma is to assist the optimization of pharmacological treatment as well as the development of potential new target-guided therapy [4]. Pharmacogenomics may play a key role in the development of therapeutic options that better suit each individual’s unique needs.

Treatment response for asthma can be assessed through different clinical parameters, such as lung function measurements, quality of life, asthma symptoms and exacerbations, but also by other indicators of failure of the treatment, such as: (1) hospital admissions for asthma, and (2) a course of oral corticosteroids or (3) emergency room visits because of asthma. Moreover, several factors can influence poor outcomes—among them, age, continued exposure to allergens and pollutants, and genetic and epigenetic factors.

Herein, we provide a concise update of genetic determinants of poor response to treatment in severe asthma, and future directions reviewing the latest publications in the field. For the sake of this review, we have considered the phenotypes identified by lung function measurements and exacerbation despite treatment as a marker of a favorable or poor response to treatment in asthma.

## 2. Pharmacogenomics of the Therapeutic Response of Asthma

It is well-known that the therapeutic response in asthma is heterogeneous, which can be at least partially explained by interindividual variability. Over the past few years, researchers have evaluated these associations through hypothesis-free strategies, such as the Genome-Wide Association Study (GWAS), which has uncovered novel pharmacogenetic variants related to inhaled corticosteroids (ICS) and bronchodilator responses.

Samedy-Bates and collaborators [5] evaluated the association between ICS use and the bronchodilator response (BDR) in three pediatric populations with persistent asthma (African American, Puerto Rican, and Mexican American children). ICS treatment was significantly associated with increased BDR among Mexican Americans. However, no difference was found in the other two populations, indicating that the benefit of using ICS may vary between racial/ethnic groups, as a consequence of the genetic difference between populations.

A systematic review published in 2019 analyzed a subset of seven GWAS on the inhaled corticosteroid response [8]. It reports that few genes’ effects have been replicated in GWAS or even in candidate gene studies. Such an observation can be explained by lack of harmonization between the analyzed phenotypes or even the genetic architecture of the studied populations. According to the systematic review, the most replicated variants were rs242941 and rs1876828 in CRHR1 [9], the rs37973 in GLCCI1 [9,10,11,12], the rs28364072 in FCER2 [12,13,14,15,16], the rs2240017 in TBX21 [9,12,13,14,15,16], the rs41423247 in NR3C1 [9], and variants in the 17q21 locus [17].

In the period considered for the present review, only three GWAS were published and may contribute to new loci associated with a poor response to inhaled corticosteroids. In the first, Hernandez–Pacheco et al. [18] carried out a meta-analysis involving two GWAS of asthma exacerbations in Hispanic and African American children treated with ICS. No variant was associated with the level of statistical significance required for a GWAS. However, 11 variants were suggestively associated with asthma exacerbations. Of these, only the variant rs5995653, at the intergenic region of APOBEC3B and APOBEC3C, was replicated in another population studied.

A second GWAS carried out by the same group evaluated asthma exacerbations in a different population, including children of European descent treated with ICS from eight studies [19]. Ten variants were associated with asthma exacerbations in patients treated with ICS. Moreover, one variant, rs67026078, located within the intergenic region of *CACNA2D3* and *WNT5A,* have been replicated among Europeans, but not validated in non-European populations.

The same group performed a combined analysis of transcriptomic and genetic data in relation to treatment responses among individuals with asthma. The LTBP1 was the most consistently associated gene in a favorable ICS response among asthma patients. Analysis of this gene revealed the rs11681246 was negatively associated with asthma exacerbations regardless of ICS use in Europeans but not in non-Europeans. Conversely, the rs76390075, also located in LTBP1, was negatively associated with the same outcome in the non-European population (Table 1) [20].

Similar to the response to ICS, the bronchodilator drug response (BDR) also varies among population groups as a consequence of the genetic variability. As demonstrated by Spear and colleagues [21] in African American children with asthma, the rs73650726, located in the non-coding RNA LOC105376110, was negatively associated to BDR with statistical significance at the GWAS (Table 1). Additionally, the other three single-nucleotide polymorphisms (SNPs) in the PRKG1 (Protein Kinase, CGMP-Dependent, Type I) were associated with the BDR in the meta-analysis across African Americans and Latin Americans living in the USA (rs7903366, rs7070958, and rs7081864).

Recently, a meta-analysis of Genome-Wide Association Studies (meta-GWAS) of asthma exacerbations on regular long-acting beta 2 agonist (LABA) use was performed in children and young adults with asthma from six studies. No variant was associated with the outcome with the GWAS significance level. However, eight variants were suggestively associated with a poor response to regular LABA treatment (p-value threshold ≤ 5 × 10^−6^), including variants in two loci (*TBX3* and *EPHA7)* previously implicated in the response to short-acting beta2-agonists (SABA) [22]. Further studies in larger populations with a diverse genetic structure will be required to validate these results.

## 3. Candidate Genes Studies Linked to Therapeutic Response of Asthma

### 3.1. Influence of Genetic Variant in ADRB2 on Response to the Bronchodilator

A systematic review has reported that the ADRB2 encoding the beta2 adrenergic receptor is the most-studied gene on the pharmacogenetics of the bronchodilator response [23]. Several studies have found associations with three non-synonymous genetic variants, rs1042713 (Arg16Gly), rs1042714 (Gln27Glu), and rs180888 (Thr164Ile) with the BDR in children and adults with asthma [24,25,26,27,28,29] although these results are still controversial [30,31,32,33] (Table 1).

In this intricate context, Hikino et al. [34] conducted a meta-analysis involving seven studies to evaluate the effects of the rs1042713 and rs1042714 on Forced Expiratory Volume in 1 s as a proportion of the predicted (FEV_1_%) after albuterol use in asthma patients. No difference in FEV_1_% between the genotypes of either SNP was found. However, in subgroup analyses, significant associations were found for rs1042713 in studies where no methacholine bronchoconstriction was conducted and among studies that included patients with no co-morbidities. These observations demonstrate the inconclusive impact of variants on ADRB2 on the bronchodilator response, highlighting the need for further studies (Table 1).

Additionally, a study conducted by Toraih et al. [35] in an Egyptian population evaluated both variants related to the bronchodilator response. Although it was published prior to the meta-analysis described above, it was not included in the study carried out by Hikino et al. [34]. The authors found the carriers of the rs1042713/rs1042714 (A/C; Arg16/Gln27) haplotype exhibited a better response to treatment, which makes it seem that the two variants are important and need to be investigated together. This finding may explain, at least in part, the lack of association found in the meta-analysis, which evaluated the variants individually (Table 1).

A study carried out by Bhosale et al. [36] investigated the molecular impact of the rs1042714 variant on the structure of the β2AR and binding of the beta agonists albuterol, isoproterenol, and terbutaline. It was observed that the G allele (Glu variant) alters electrostatic interactions in the region of the binding site which results in better bonding with the three ligands tested compared to the wild type, which can explain the variable therapeutic response to adrenergic agonists.

Despite the large number of studies with variants in ADRB2, to this day it is not possible to use them as biomarkers of the response to bronchodilators. To evaluate the clinical application of these variants, the PharmGKB^®^ (https://www.pharmgkb.org/ accessed 13 Mar 2021) was consulted. This is a NIH-funded resource managed at Stanford University that provides information about how genetic variants affect the therapeutic response, assigning levels from 1–4, with level 1 meeting the highest criteria [37]. Among the pharmacogenetic variants for the therapeutic response of asthma, the variant rs1042713 has the highest score on the PharmGKB^®^ dataset, with a 2A level for the efficacy of salbutamol and salmeterol. Future studies, including intervention studies, should help to characterize its functional importance, contributing to implementation of this marker in clinical practice.

### 3.2. Variants in GLCCI1 Associated with Response to Corticosteroids

Current guidelines recommend ICS as first-line therapy in persistent asthma as its anti-inflammatory effect reduces airway inflammation and risk of exacerbations, and improves disease control [38]. Despite the proven clinical efficacy of ICS, the intraindividual response is variable and likely influenced by genetic background and environmental exposures [39]. GLCCI1 is involved in inflammatory cell apoptosis and its expression is induced by glucocorticoids [40].

A GWAS carried out by Tantisira and colleagues [41] has shown that genetic variations in glucocorticoid-induced transcript 1 genes (GLCCI1) are linked with decreased lung function in subjects treated with ICS. The single nucleotide variant (SNV) rs37972 was associated with changes in FEV_1_ and marked attenuation of the response to ICS treatment. Additionally, the rs37973 SNP, which down-regulates GLCCI1 expression, was related to significantly reduced lung function and altered clinical responsiveness to ICS. Homozygous carriers of a mutant allele (TT) were about two-and-a-half times more likely to have an impaired ICS response compared to homozygous ones for the wild-type allele (CC). In a Japanese population with asthma under long-term ICS treatment, rs37973 SNP was also associated with response to ICS treatment and FEV_1_ decline of 30 mL/year or greater [40]. Conversely, however, a Slovenian study reported that rs37973 SNP was positively associated with ICS responsiveness, even though it was considerably modified by smoking and atopy [42] (Table 1).

The association between genetic variants and exacerbation risk was recently addressed in the largest pharmacogenetic candidate gene study to date in the Dutch population-based Rotterdam study with replication in the American GERA cohort [38]. Single-nucleotide polymorphisms for GLCCI1 (rs37973) significantly increased exacerbation risk in patients treated with ICS in the Dutch cohort, but an opposite effect was seen in the American replication cohort. The rs37973 is level 3 on PharmGKB^®^, which means that further research is required for a better understanding of the implication on asthma exacerbations.

### 3.3. Genetic Variants in the Vitamin D Pathway and the Therapeutic Response on Asthma

The active metabolite of vitamin D, 1α-25-dihydroxyvitamin D, has a substantial immunomodulatory influence on the expression of proinflammatory transcriptional regulators, the inflammatory cellular response, cytokines, and IgE regulation. Variants in the vitamin D pathway gene (VDR, CYP2R1, CYP24A1, CYP27B1, DBP) are related, typically, to molecular pathways that affect the phenotype of asthma [43].

A study with steroid-resistant asthma patients demonstrated that the oral treatment with calcitriol, the active form of vitamin D, augments IL-10 production and reduces IL-17A production from peripheral blood mononuclear cells after exposure to dexamethasone, when compared to patients with steroid-sensitive asthma [44]. Additionally, budesonide and calcitriol up-regulate the expression of VDR and Smad7 and inhibit the production of proteins involved in airway remodeling. These effects are enhanced when both (budesonide and calcitriol) are used in combination [45]. Together, these findings highlight the likely importance of the vitamin D pathway in the response to corticosteroids.

Mohamed and collaborators [46] evaluated the association between variants in the Vitamin D receptor (VDR) gene, and the response to glucocorticoids revealed a three times greater risk of individuals with the T allele of the rs2228570 (FokI) variant to be resistant to glucocorticoids. In another study evaluating the same variant, children with asthma carrying the polymorphic allele had higher exacerbation severity scores and a poorer β_2_-agonist treatment response [47] (Table 1).

## 4. Interaction between Genetics and Age in Response to Asthma Treatment

An aspect seldom addressed in genetic studies is the impact of age on genetic predisposition. In this context, a study by Dahlin and colleagues [48] performed a genome-wide interaction study (GWIS) to evaluate, for the first time, the interaction of genetic variation and age on ICS response (measured by the occurrence of exacerbations) on 1321 adults and children of European ancestry with asthma. The top SNP was the rs34631960 in THSD4 (thrombospondin type 1 domain containing protein 4) that was associated with risk of poor ICS response with increasing age (P < 5 × 10^−8^). Moreover, the rs2328386 in HIVEP2 (human immunodeficiency virus type I enhancer binding protein 2) was a protective factor for the lack of response to ICS with increasing age. Another 107 age-by-genotype interaction met genome-wide suggestive significance.

A similar analysis was carried out between the interaction of genetic factors and age on the response to the bronchodilator from childhood to adulthood with asthma. Sordillo et al. [49] demonstrated for the first time that the rs295137 and the rs2626393 had their association with the BDR modified by age. In early childhood through to adolescence, the rs295137 variant (T allele) near the SPATS2L gene was associated with an increased BDR response, while the rs2626393 variant (C allele) near the ASB3 gene was associated with lower BDR with a significant decrease in the magnitude of effect from adolescence to adulthood.

These findings can be explained by the impact of the accumulation of environmental exposures on the genetic predisposition, determined by epigenetic changes.

## 5. Epigenetic Mechanisms Involved in the Lack of Therapeutic Control of Asthma

Epigenetic mechanisms mediate nonstructural changes in gene expression that are regulated by DNA methylation patterns, microRNA expression, histone modifications, or a variety of other mechanisms [50]. These changes in gene expression patterns might be related to environmental exposure over time, since increasing genetic divergence in monozygotic twins has been described with advancing age [4].

MicroRNAs (miRNAs) are small noncoding RNA molecules that act as a regulator of gene transcription. Recent studies have demonstrated the role of miRNAs in the therapeutic response of patients with asthma which reinforces the potential for their future use as a predictor biomarker of treatment outcomes. Li et al. [51] identified seven miRNAs (hsa-miR-155-5p, hsa-miR4433b-5p, hsa-miR-532-5p, hsa-miR-345-5p, hsa-miR-652-3p, hsa-miR-126-3p, and hsa-miR-335-5p) associated with lung function changes over 4 years on ICS (budesonide) treatment. Improvement of lung function linked to a better ICS response was significantly associated with hsa-miR-155-5p, hsa-miR4433b-5p, hsa-miR-126-3p, and hsa-miR-335-5p miRNAs, while hsa-miR-532-5p, hsa-miR-345-5p, and hsa-miR-652-3p were linked to relative decrease in FEV1%. Additionally, the hsa-miR-155-5p and hsa-miR-532-5p were functionally validated on dexamethasone-related NF-κB transrepression, and together were predictive of ICS response [47]. Another microRNA with potential for therapeutic prediction is the miR-16, which inhibits the expression of ADRB2, has been negatively correlated with FEV_1_ and may function as a biomarker of the response to salmeterol therapy [52].

The DNA methylation is an epigenetic process related to the addition of a methyl group to a cytosine residue in a CpG site and has also been shown to interfere with the therapeutic outcome of asthma. Wang and colleagues [53] have evaluated the association between blood DNA methylation and ICS response in children with persistent asthma measured by the percentage change in FEV_1_ eight weeks after treatment initiation. In those individuals, hypermethylation of cg27254601 was associated with both a response to ICS and BOLA2 expression. Additionally, 551 CpG sites were also differentially methylated. The blood ADRB2 5′ untranslated region (5′-UTR) methylation level was also positive associated with a risk of uncontrolled asthma in children. This methylation level was positively correlated with whole blood aluminum concentration in asthmatic children, highlighting the potential that environmental exposures have to modify epigenetic factors [54].

Together, these findings indicate that the lack of response to asthma treatment is a consequence not only of genetic predisposition, but also of environmental/epigenetic factors. Future studies should be carried out to investigate whether environmental exposures are efficient to reprogram genetic predisposition through changes in epigenetic patterns.

## 6. Conclusions

Poor control of severe asthma is a complex multifaceted outcome. It has been related to an increasing number of genetic variants and epigenetic changes, whether in genes of pharmacological pathways, or in genes involved in the pathophysiology of asthma. There is still no genetic variant that can be used as a biomarker to predict the therapeutic response in patients with asthma, although the LTBP1 was the most consistently associated gene in a favorable ICS response among asthma patients, and genetic variations in glucocorticoid-induced transcript 1 genes (GLCCI1) were linked with decreased lung function in subjects treated with ICS.

ADRB2 encoding the beta2 adrenergic receptor is the most-studied gene on the pharmacogenetics of bronchodilator response; however, its impact in clinical practice is still controversial. According to PharmGKB, only two of the variants described in the recent studies are more likely to have a pharmacogenetic impact on asthma. The variant rs1042713 reached the highest score on the PharmGKB^®^ dataset, with a 2A level for the efficacy of salbutamol and salmeterol.

The variant rs67026078, located within the intergenic region between *CACNA2D3* and *WNT5A,* was associated with asthma exacerbations in children. These findings have been replicated among Europeans, but not validated in non-European populations.

Future studies should be dedicated not only to identify new variants, but also to validate the associations previously described, such as a clinical intervention strategy based on the variants of interest. The implementation of pharmacogenetic algorithms for asthma management might contribute to a more effective and safe treatment for the patient through therapeutic personalization.

## Figures and Tables

**Figure 1 ijms-22-04251-f001:**
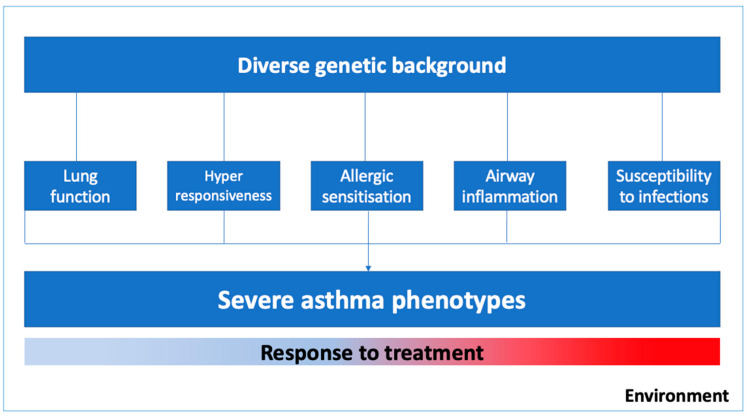
Environmental factors and biogenetic mechanisms related to severe asthma phenotypes.

**Table 1 ijms-22-04251-t001:** Characterization of the single-nucleotide polymorphisms (SNPs) described in recent pharmacogenetic studies of the therapeutic response to bronchodilators or inhaled corticosteroids in asthma.

SNP	Score PharmGKB	Gene	Outcome	Reference
rs11681246	-	LTBP1	Negatively associated with asthma exacerbations regardless of ICS use	[20]
rs76390075	-	LTBP1		[20]
rs73650726	-	LOC105376110	Negatively associated with BDR	[21]
rs7903366	-	PRKG1	Positively associated with the BDR	[21]
rs7081864	-	PRKG1	Positively associated with the BDR	[21]
rs7070958	-	PRKG1	Negatively associated with the BDR	[21]
rs1042713	2A	ADRB2	Better response to treatment with A allele	[24,30]
			Worst response to treatment with A allele	[29]
			No association with therapeutic response	[25,32,34]
rs1042714	-	ADRB2	Better response to treatment	[23,29]
			No association with therapeutic response	[24,27,30,34]
rs1042713/rs1042714	-	ADRB2	Better response to treatment	[35]
rs180888	-	ADRB2	Uncontrolled asthma during LABA treatment	[28]
rs37972	-	GLCCI1	Attenuation of the response to ICS treatment	[41]
rs37973	3	GLCCI1	Worst response to treatment with G Allele	[38,40,41]
			Better response to treatment with G allele	[42]
rs2228570	-	VDR	Risk of resistance to inhaled glucocorticoids	[46]
			Higher exacerbation severity scores and poorer β_2_-agonist treatment response	[47]

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
