# Peer review of "Genetic Determinants of Poor Response to Treatment in Severe Asthma"

_ijms, 2021, doi:10.3390/ijms22084251_

Round 1

Reviewer 1 Report

This review on the genetic and epigenetic determinants of asthma is very interesting. Asthma is a multidisciplinary disease and unfortunately, as pointed out by the Authors, many patients are still not well controlled by the currently available drug therapy.

I have just few suggestions:

  1. line 286: it is to be put FEV1 in place of VEF1, wrongly written;
  2. lines 353-355: it is not clear what the relationship is with the lung function data, it is only reported "...FEV1 % change...".  I suggest explaining this point better 
  3. The paper reports the various genetic and epigenetic evidences. In lines 373-376 comments of the Authors are reported, however in an excessively concise manner. I suggest inserting a discussion paragraph in which the authors summarize and underline the strengths and weaknesses of what they have previously reported. This paragraph would help the reader and make the article clearer 

Author Response

We are grateful to the reviewer for their insightful comments on our paper. We have been able to incorporate changes to reflect most of the suggestions provided by the reviewer. Here is a point-by-point response to the reviewers' comments and concerns.

Point 1: line 286: it is to be put FEV1 in place of VEF1, wrongly written;

Response 1: Revised accordingly.

Point 2: lines 353-355: it is not clear what the relationship is with the lung function data, it is only reported "...FEV1 % change...".  I suggest explaining this point better  

Response 2: We have revised the text in order to make our point clearer connecting lung function changes and miRNAs, as follows:  Li and collaborator [51] identified 7 miRNAs (hsa-miR-155-5p, hsa-miR4433b-5p, hsa-miR-532-5p, hsa-miR-345- 5p, hsa-miR-652-3p, hsa-miR-126-3p, and hsa-miR-335-5p) associated with lung function changes over 4-years time on ICS (budesonide) treatment. Improvement of lung function linked to a better ICS response was significantly associated with hsa-miR-155-5p, hsa-miR4433b-5p, hsa-miR-126-3p, and hsa-miR-335-5p miRNAs, while hsa-miR-532-5p, hsa-miR-345- 5p, hsa-miR-652-3p were linked to relative decrease in FEV1%. [lines 525-536].

Point 3: The paper reports the various genetic and epigenetic evidences. In lines 373-376 comments of the Authors are reported, however in an excessively concise manner. I suggest inserting a discussion paragraph in which the authors summarize and underline the strengths and weaknesses of what they have previously reported. This paragraph would help the reader and make the article clearer

Response 3: We agree with the reviewer that more quantitative information would add relevance for the reader. We have revised the text which we present below:

“Although the LTBP1 was the most consistently associated gene in a favorable ICS response among asthma patients and genetic variations in glucocorticoid-induced transcript 1 genes (GLCCI1) were linked with decreased lung function in subjects treated with ICS.

ADRB2 encoding the beta2 adrenergic receptor is the most studied gene on the pharmacogenetics of bronchodilator response, however, its impact in clinical practice is still controversial. According to PharmGKB, only two of the variants described in the recent studies are more likely to have a pharmacogenetic impact on asthma. The variant rs1042713 reached the highest score on the PharmGKB® dataset, with a 2A level for the efficacy of salbutamol and salmeterol. 

The variant rs67026078, located within the intergenic region between CACNA2D3 and WNT5A, was associated with asthma exacerbations in children. These findings have been replicated among Europeans, but not validated in non-European populations.”    [lines 562-575].

Reviewer 2 Report

IJMS

Manuscript ID: 1185287

Type of manuscript: Review

Title: Genetic determinants of poor response to treatment in severe asthma

This review is quite nice compact update of information on genetic determinants of poor response to treatment of severe asthma. The article includes consideration on pharmacogenomics of the therapeutic response of asthma; candidate genes studies related to therapeutic response of asthma, interaction between genetics and age and role of epigenetic mechanisms engaged in the lack of therapeutic control of asthma.

I have several minor remarks that need to be addressed:

- line 145-146: “early life viral lower respiratory infections”. Could you please rewrite this, because the reader can’t be sure what you meant, lower level of early life viral infections or infections of lower respiratory tract?

- line 147: “sensibilization”. Did you mean sensitization?

- line 159-161: This sentence is quite confusing, please rewrite this: “The potential for personalized medicine and pharmacogenetics inform the choosing wisely therapeutical options that better suit each individual´s unique needs is compelling.”

- line 167: Shouldn’t it be: “such as continued exposure…”

- line 198: “admixed children”. What did authors mean? Please specify what admixed mean in this context.

- “SNP” and “SNV”, please define this in Table 1 and in the text when it appears for the first time.

- line 286: VEF1- what is that? Did you mean FEV1, otherwise please define.

- line 311: “PBMC”- please define or isn’t it better use full name since this appears in the text only once

-line 384: Shouldn’t it be: “…be dedicated not only to identify new variants….”

Author Response

We are grateful to the reviewer for their insightful comments on our paper. We have been able to incorporate changes to reflect most of the suggestions provided by the reviewer. Here is a point-by-point response to the reviewers' comments and concerns.

Point 1: line 145-146: “early life viral lower respiratory infections”. Could you please rewrite this, because the reader can’t be sure what you meant, lower level of early life viral infections or infections of lower respiratory tract?

Response 1: Modified in the abstract to: “lower respiratory tract infections in early life”

Point 2: line 147: “sensibilization”. Did you mean sensitization? 

Response 2: Revised accordingly.

Point 3: line 159-161: This sentence is quite confusing, please rewrite this: “The potential for personalized medicine and pharmacogenetics inform the choosing wisely therapeutical options that better suit each individual´s unique needs is compelling.”

Response 3: The sentence now reads Pharmacogenomics may play a key role in the development of therapeutic options that better suit each individual´s unique needs” [lines 161-163].

Point 4:  line 167: Shouldn’t it be: “such as continued exposure…”

Response 4: We appreciate this recommendation. The sentence now reads Moreover, several factors can influence poor outcomes, among them, age, continued exposure to allergens and pollutants, genetic and epigenetic.” [lines 168-169].

Point 5:  line 198: “admixed children”. What did authors mean? Please specify what admixed mean in this context.

 Response 5: We have clarify this point in the text as follows: “In the first, Hernandez-Pacheco et al. [18] carried out a meta-analysis involving two GWAS of asthma exacerbations in Hispanics and African Americans children treated with ICS” [lines 210-212]. 

Point 6: “SNP” and “SNV”, please define this in Table 1 and in the text when it appears for the first time.

Response 6: We have revised the paper accordingly. 

Point 7:  Line 286: VEF1- what is that? Did you mean FEV1, otherwise please define.

Response 7: Yes, the correct is FEV1. We have properly corrected that.

Point 8: Line 311: “PBMC”- please define or isn’t it better use full name since this appears in the text only once

Response 8: We agree with the reviewer that it is better to use the full name “peripheral blood mononuclear cells”.

Point 9: Line 384: Shouldn’t it be: “…be dedicated not only to identify new variants….”

Response 9: Revised accordingly.
